# Overnutrition of Ewe in Late Gestation and the Impact on Placental Efficiency and Lamb’s Performance

**DOI:** 10.3390/ani13010103

**Published:** 2022-12-27

**Authors:** Marco Antonio Paula de Sousa, Sergio Novita Esteves, Luciara Celi Chaves Daher, Sarita Bonagurio Gallo, Verônica Schinaider do Amaral Pereira, Jeferson Ferreira da Fonseca, André Guimarães Maciel e Silva, Felipe Zandonadi Brandão, Maria Emilia Franco Oliveira, Andréa do Nascimento Barreto, Gabriel Brun Vergani, Alexandre Rossetto Garcia

**Affiliations:** 1Institute of Veterinary Medicine, Federal University of Pará, Av. dos Universitários, s/n, Castanhal 68746-360, PA, Brazil; 2Brazilian Agricultural Research Corporation-Embrapa Southeast Livestock, Rod Washington Luiz, km 234, São Carlos 13560-970, SP, Brazil; 3Institute of Animal Health and Production, Federal Rural University of Amazon, Av. Presidente Tancredo Neves, 2501, Belém 66077-830, PA, Brazil; 4Faculty of Animal Science and Food Engineering, University of São Paulo, Av. Duque de Caxias Norte, 225, Pirassununga 13635-900, SP, Brazil; 5Brazilian Agricultural Research Corporation-Embrapa Goats and Sheep, Estrada Sobral/Groaíras, km 4, Sobral 62010-970, CE, Brazil; 6School of Veterinary Medicine, Fluminense Federal University, Rua Vital Brazil, 64, Niterói 24230-340, RJ, Brazil; 7Department of Pathology, Reproduction and One Health, School of Agricultural and Veterinarian Sciences, São Paulo State University, Via de Acesso Prof. Paulo Donato Castellane, s/n, Jaboticabal 14884-900, SP, Brazil

**Keywords:** maternal nutrition, fetal development, fetal programming, placenta, neonatal behavior, Morada Nova

## Abstract

**Simple Summary:**

Ewe nutrition during gestation, either in excess or in deficit, may influence fetal development and the postnatal life of the lamb. The objective of this study was to evaluate the effects of use the of energy (ME) and energy/protein (MECP) supplementation in the ewe’s diet in the last third of gestation on maternal placental and endocrine characteristics, as well as on the behavior of neonatal lambs and productive performance. ME and MECP ewes had higher serum concentrations of triiodothyronine and cortisol on the day of lambing. However, only ME ewes had higher placental efficiency and lower total cotyledon weight. The ME and MECP diets increased birth weight at weaning and reduced the time to kneel, to try to stand, and the latency to stand of the lambs. It is concluded that overnutrition in the last third of gestation was positive for the sheep production system, with improved production rates, hormonal profile, placental characteristics and neonatal behavior. These results are valuable to society because they indicate a management strategy during the reproductive season that can be adopted by sheep producers to increase the productivity of their flocks.

**Abstract:**

The objective of the study was to evaluate energy or energy/protein supplementation in the ewe diet, in the last third of gestation, on maternal placental and endocrine characteristics, as well as its effects on the behavior of neonatal lambs and productive performance until weaning. A total of 128 ewes were used, and the experimental diet was fed from 100 days gestation until lambing, with the birth of 172 lambs. The ewes were distributed into three treatments: CTL (control, *n* = 43) with metabolizable energy (ME) and crude protein (CP) intake according to NRC (1985); ME (energy supplementation, *n* = 44) plus 21% ME; and MECP (energy/protein supplementation, *n* = 41) plus 26% ME and CP. Body weight, body condition score, serum hormone concentrations, placental characteristics, lamb performance and behavior, and production efficiency of the ewe from lambing to weaning were measured. ME and MECP ewes were heavier before (*p =* 0.006) and just after lambing (*p =* 0.002) and had higher serum triiodothyronine (*p =* 0.001) and cortisol (*p =* 0.004) concentrations on the day of lambing. ME ewes had higher placental efficiency (*p =* 0.036) and lower total cotyledon weight (*p =* 0.011). ME and MECP diets increased both lamb birth weight (*p =* 0.015) and weaning weight (*p =* 0.009). Production efficiency at birth and at weaning was not influenced (*p >* 0.05) by treatments. Lamb behavior was influenced by the ME and MECP diets, reducing the time to kneel (*p ≤* 0.05), to try to stand (*p ≤* 0.05), and the latency to stand (*p ≤* 0.005). It is concluded that overnutrition in the last third of gestation was positive for the ovine production system, with improved production rates, hormonal profile, placental characteristics, and neonatal behavior.

## 1. Introduction

The growing appreciation and demand for sheep meat, especially from young animals, has stimulated the intensification of production systems. This has made the development of new technologies to increase lamb performance essential, including for such breeds as Morada Nova. The Morada Nova is an indigenous hair sheep breed developed in Brazil with a short and silky hair coat [1]. The Morada Nova is a medium-frame pooled breed presenting a coat color of various shades of red. The average body height of mature ewes is around 61 cm and around 65 cm for the rams [2]. Ewe nutrition during gestation, either excess or deficiency, can influence fetal development and postnatal life, altering the lamb’s growth and physiological functions and compromising its productive performance [3,4].

The placenta is the organ that performs the main interface between mother and fetus and plays an essential role in maintaining fetal growth and development by facilitating the transfer of oxygen and nutrients [5]. Placental efficiency, defined by [6] as the grams of fetus produced per gram of placenta, is a direct consequence of the placenta’s ability to transfer nutrients to the fetus [7].

During late gestation, the energy demands of ewes rise due to the increased nutritional requirements and the availability of nutrients for fetal growth. During neonatal life, glucose is the main energy substrate used in fetal and placental tissue metabolism [8]. In late gestation, in addition to fetal growth, there is evidence that the nutritional status of the gestating ewe also impacts neonatal behavior [9]. Ideally, neonatal lambs should be vigorous to quickly locate the maternal mammary glands, which improves the chance of survival in the early postpartum period and enhances the development until weaning [10].

This experiment hypothesizes that increased energy and/or energy-protein intake in the nutrition of the pregnant ewe will benefit the ewe’s body reserve, placental efficiency, and the lambs’ body weight and behavior. Therefore, the objective of the study was to evaluate the influence of energy or protein/energy supplementation in ewe nutrition during the last third of gestation on placental morphometric and maternal endocrine characteristics, as well as its effects on the behavioral expression of lambs in the immediate postpartum period and on their productive performance at weaning.

## 2. Material and Methods

### 2.1. Ethics Statement

The study was conducted following the Brazilian Guideline for the Care and Use of Animals in Teaching or Scientific Research Activities [11]. The experimental procedures were previously approved by the Ethics Committee on Animal Use of Embrapa Pecuária Sudeste (Protocol 09/2017) and reported according to The ARRIVE Guidelines: Animal Research: Reporting of In Vivo Experiments [12].

### 2.2. Location and Animals

Utilizing 134 Morada Nova hair sheep ewes (3.9 ± 1.5 years), the experiment was conducted at Embrapa Pecuária Sudeste, São Carlos, Brazil. The ewes had their reproductive cycle synchronized (60 mg medroxyprogesterone intravaginal sponge for six days plus 200 IU eCG i.m., 30 μg d-cloprostenol i.m. 36 h before sponge removal), and after estrus detection, natural mating was performed with rams of the same breed, with libido and seminal quality previously evaluated (5 to 8 ewes/ram). Each ewe received two mates at intervals of 12 h, and gestation was checked 30 days later by real-time ultrasonography (DP-3300Vet, Mindray Bio-Medical Electronics Ltd., Shenzhen, China) using a 5–7.5 MHz transrectal linear transducer. A total of 128 ewes were confirmed pregnant and utilized for the experiment through lambing, with 172 lambs resulting for use in the study. Only ewes with twin pregnancies of two lambs were used in the experiment, and a similar number of sire pregnancies (rams) was assigned to each treatment.

The ewes were kept together on pasture of Aruana grass (*Panicum maximum* cv. Aruana) until reaching 100 days of gestation. The ewes had a body weight (BW) of 41.0 ± 0.5 kg and a body condition score (BCS) of 3.0 ± 0.5 on a scale of 1–5 [13]. From this point on, the ewes were divided into three treatments in an entirely randomized design regarding age and lambing order.

### 2.3. Experimental Treatments

Experimental treatments were CTL (control, *n* = 43 ewes) with metabolizable energy (ME) and crude protein (CP) intake according to [14], ME (energy supplementation, *n* = 44 ewes) diet with 21% more ME only, and MECP (energy/protein supplementation, *n* = 41 ewes) diet with 26% more ME and CP. The animals in each treatment occupied an area of 0.8 ha, divided into four 0.2 ha paddocks for intensive rotational grazing. The sheep had ad libitum access to water, mineral mixture, and shade. The concentrated diet was formulated for each treatment (Table 1), and the feedstuffs used were ground corn, soybean meal, and vegetable oil. ME and MECP were fed twice daily at 8 a.m. and 5 p.m., while CTL received the full diet at 8 a.m.

The chemical composition of the experimental diets was analyzed for dry matter (DM), crude protein (CP), ether extract (EE), neutral detergent fiber (NDF), and acid detergent fiber (ADF), according to techniques described by [15]. The amount of total digestible nutrients (TDN = 74.49 − 0.5635 × ADF) was calculated according to [16]. Thereafter, to calculate the digestible energy (DE), 1 kg of TDN was considered equivalent to 4.409 Mcal, and to calculate the metabolizable energy the DE was multiplied by 0.82, according to the [14].


The dry matter intake of the animal was estimated using Equation (1).
***BW at lambing* = (*adjusted BW* + *BW at* 100 *days of gestation*)/2**(1)
where:

BW = body weight, kg

Adjusted BW, kg = ewe’s weight up to 12 h after parturition (kg) + lamb´s weight at birth (kg) + weight of the placenta (kg).

The forage and concentrate intake for CTL was 0.67 and 0.43 kg, for ME it was 0.60 and 0.52 kg, and for MECP it was 0.43 and 0.71 kg per animal per day, respectively.

**Table 1 animals-13-00103-t001:** Nutritional composition and estimated daily intake of experimental diets offered to ewes submitted or not submitted to overnutrition in the last third of gestation (CTL—control; ME—energy supplementation; MECP—energy/protein supplementation).

*Ingredient*, %	Fodder	Treatments		
		CTL	ME	MECP
Soybean meal		3.1	10.3	17.4
Ground corn grain		95.1	75.2	72.5
Soybean oil		0.7	13.2	8.5
Calcite limestone		1.1	1.3	1.6
Monensin sodium		0.035	0.040	0.040
***Nutritional Composition***, %				
Dry Matter	23.89	89.19	90.44	90.90
Crude Protein	11.04	10.71	11.88	14.74
Ether Extract	2.20	4.39	16.09	11.58
Neutral detergent fiber	72.80	15.11	13.52	14.46
Acid detergent fiber	42.10	3.88	4.01	4.77
Total digestible nutrients	50.77	72.40	93.39	84.95
*Dry matter intake, kg*/*d*^¥^		1.10	1.12	1.14
Nutrient Intake, %				
Crude Protein		12.00	12.00	15.21
Ether Extract		3.36	9.68	9.16
Neutral detergent fiber		55.27	50.71	41.57
Acid detergent fiber		29.87	27.34	21.48
Total digestible nutrients ^⸶^		65.14	79.02	82.14
Digestible energy, *Mcal* ^£^		2872.37	3483.99	3619.33
Metabolizable energy, *Mcal* ^Δ^		2355.34	2856.87	2967.85

^¥^ Calculated on the basis of the average live weight at birth (average value between the adjusted weight at birth and at 100 days of gestation); ^⸶^ Estimated value NDT = 74.49 − 0.5635 × FDA [16]; ^£^ Digestible Energy = 1 kg TDN equals 4.409 Mcal [14]; ^Δ^ Metabolizable Energy (ME = 0.82 × DE; [14]).

### 2.4. Lambing Monitoring

Ten days before the expected lambing date, the ewes were continuously observed until lambing, being kept in pasture during the day (6 a.m. to 6 p.m.) and in a covered shed during the night and early morning (6 p.m. to 6 a.m.). The ewes remained under uninterrupted visual monitoring, performed in rotation by four previously trained observers. Once detected in the prodromal phase of parturition, when the ewes present inappetence, seek isolation from the group, and show anxiety and agitation [17], the ewes were monitored for progressive signs of labor. The interval of sixty minutes was adopted as the maximum time for the normal progression of labor. Otherwise, veterinary medical assistance would be immediately called for obstetric assistance, which was not necessary for any of the sheep. Once lambing occurred, the observation of their respective lambs was immediately initiated, so that all lambs born from the experimental ewes were evaluated.

### 2.5. Ewe’s Performance

The ewes were weighed within 12 h of lambing and at 90 days (age at weaning), under a 12 h feed-fasting period in a mobile mechanical balance, cage-type scales with a capacity of 300 kg and precision of 0.1 kg. The weight of the ewe at lambing was calculated as described in Equation (1). Body condition was evaluated using a scale of 1–5 [13], being considered as 1-emaciated, 2-thin, 3-average, 4-fat, 5-obese. The lambs were weighed individually after birth and at weaning (90 days) with the aid of a portable digital scale with a maximum capacity of 50 kg and precision of 1 g, equipped with accessory support proper for weighing lambs.

### 2.6. Hormone Concentration

Blood samples from the ewes were collected by venipuncture in vacuum tubes without anticoagulant (Vacutainer, New Jersey, USA) immediately after lambing. The samples were centrifuged at 1350× *g* for 15 min for complete serum separation, fractionated in polypropylene microtubes, and stored at −20 °C until analysis. The samples were evaluated for cortisol and triiodothyronine (T3) concentrations by radioimmunoassay. Cortisol was measured with the Cortisol *Immuchem Coated Tube kit* (MP Biomedicals, LCC Diagnostics Division, Santa Ana, CA, USA). Levels of T3 were measured with T3 Antibody-Coated Tubes, T3 Tracer [125I], and T3 Standards Set (MP Biomedicals, Inc., Diagnostics Division, Santa Ana, CA, USA). The sensitivity and the intra-assay coefficient were 0.17 μg/dL and 12% for cortisol and 6.7 μg/dL and 11% for T3, respectively.

### 2.7. Placental Characteristics

During the course of labor, placentas were collected immediately after ejection and stored at 5 °C for analysis within 12 h. The placenta was sanitized and weighed on a precision digital scale, adjusted to a scale of 0.01 g. After that, the cotyledons were individually dissected, and the total number of cotyledons per placenta was measured. Subsequently, all cotyledons were evaluated for the largest diameter using a digital pachymeter (graduation of 0.05 mm), and cotyledon weight and the total cotyledon weight per placenta was calculated (grams). Placental efficiency was calculated by the formula PE = FW/PW, where FW is the weight of the lamb produced (grams) and PW is the weight of the placenta (grams), for each ewe, as proposed by [18].

### 2.8. Determination of Production Efficiency

Production efficiency at lambing and at weaning were calculated using the ratio between lamb weight at weaning and the metabolic weight of the ewe at lambing and at weaning, respectively [19,20]. The metabolic weight was calculated by raising the live weight to the power of 0.75 (BW^0.75^).

### 2.9. Neonatal Behavioral Assessment

After birth, the behavior of each lamb was directly observed (CTL = 62 lambs, ME = 59 lambs, MECP = 54 lambs). Latency times for shaking the head, getting on the knees, attempting to stand up, standing up, reaching the udder, attempting to suck, and first suck, which corresponds to the first successful suck [21,22], were used as variables of interest. The technique adopted was individual focal observation and the intervals were recorded with a manual chronometer, in minutes, for 60 min [23]. In cases of twin births, each of the lambs was treated as a focal individual, and data from their respective births and behaviors were recorded separately.

Vitality evaluation of the lambs was performed by visual assessment and the score assigned (1–5-point scale) to each lamb based on its initial postpartum behavior, with 1 being poorly active and 5 being readily active [24,25]. All behavioral observations were made by four previously trained researchers with a high degree of agreement in previous reliability tests [26].

### 2.10. Statistical Analysis

Data analysis was performed using the stats, car, lmtest, nlme, and emmeans package libraries, gamlss of R software, version 3.6.2 [27]. The explanatory variables were evaluated using a single-characteristic model fitted via generalized least squares (gls). In contrast, the cotyledon number was fitted via a generalized linear model (glm) with family = poisson. The introduction of variables to compose the final model was selected using the Stepwise method, considering the AIC as the selection criteria.

Generalized additive models for location, scale, and shape were used to analyze the variables of immediate behavior at lambing. To fit the marginal distributions for the response variables, the fitDist() function (GAMLSS package) was used. A possible nonlinear relationship was tested for the birth duration variable, adding the cubic-Spline smoothing function to the model.

Comparison between the means of the tested effects was performed by Tukey’s test. Data were reported as the adjusted mean or arithmetic mean of the sample ± standard error (SE), using the estimates (coefficients), standard errors, and *p*-value. The significance level previously adopted for the statistical analyses was ≤ 5%.

## 3. Results

### 3.1. Ewe’s Performance

Treatment promoted changes in ewe BW before (*p =* 0.006) and after lambing (*p =* 0.002), but there was no effect at weaning (*p =* 0.338). The MECP treatment resulted in ewes with higher BW both before and after lambing (49.3 and 44.9 kg, respectively), while CTL had the lightest ewes, 46.4 and 42.4 kg, respectively (Table 2). Animals from ME showed intermediate BW values when compared to the other treatments. The type of birth did not alter the BW of the ewes. However, male-pregnant ewes were heavier after lambing. The number of previous births had an effect on ewe BW (*p <* 0.0001), and ewes with 4 or more parturitions were always heavier than the others. As there was interaction between treatment and birth order at weaning, the highest BW was observed for ME ewes with four or more previous births (*p =* 0.008). The BCS of ewes did not differ between treatments (Table 2), either after lambing or at weaning, and was not influenced by type of birth, sex of offspring, or number of births (*p >* 0.05).

### 3.2. Hormone Concentration

Serum T3 and cortisol concentrations were higher (*p =* 0.001) for ME (250 ng/dL) and MECP (274 ng/dL) than for CTL (201 ng/dL, Table 3). The T3 profile was not affected by type of birth or lamb′s sex (*p >* 0.05). However, nulliparous and primiparous females showed higher T3 and cortisol release compared to ewes that had two births or more (*p =* 0.004). Cortisol concentrations after lambing were also influenced by the type of birth, with higher means for singleton ewes (*p =* 0.001).

There was a significant effect for the interaction of treatment and birth order, being observed as an increase in the concentrations of T3 and cortisol. ME and MECP promoted higher T3 release in nulliparous and primiparous ewes (*p =* 0.002). The primiparous of the MECP group also showed a significantly higher concentration of cortisol, compared to ewes with two or three deliveries (*p =* 0.001).

### 3.3. Placental Characteristics

There was no effect of maternal nutrition on placental weight (*p >* 0.05, Table 4), but twin-gestation ewes had heavier placentas (428 g) compared to single-gestation ewes (299 g, *p <* 0.0001). Offspring sex and birth order did not influence placental weight (*p >* 0.05). Placental efficiency was higher for ewes in the ME group (*p =* 0.036) compared to CTL and MECP. The type of birth and sex of the lamb also influenced placental efficiency, being higher in ewes that gestated twins (*p =* 0.019), as well as in those that delivered male lambs (*p =* 0.016). No significant effect (*p >* 0.05) of lambing order was observed on placental efficiency.

Treatments did not directly affect the total number of placental cotyledons (Table 5), but total cotyledon weight was lower for ME ewes (*p =* 0.011) relative to CTL and MECP. Twin births increased both the number and total weight of cotyledons (*p <* 0.0001), which, however, were smaller in diameter than those expressed in placentas from single lambs (*p <* 0.001). The sex of the lamb did not affect the number of cotyledons, but placentas involving male lambs showed cotyledons with a greater diameter (*p =* 0.039) and greater total weight (*p =* 0.028). Total cotyledon weight was also greater in ewes with four or more previous parturitions compared to nulliparous ewes (*p =* 0.012). The birth order influenced the diameter of cotyledons (*p =* 0.011), so primiparous ewes had lower means compared to the other ewes. In the interaction between treatment and lambing type, there was an effect for twin gestation, with a reduction in the number of cotyledons in ME ewes (*p =* 0.005). The analysis of the interaction between treatment and birth order showed that the ME treatment reduced the number of cotyledons in the nulliparous, while the MECP treatment showed this effect in females with four or more births (*p =* 0.007).

### 3.4. Lamb’s Performance

The ME (2.79 kg) and MECP (2.73 kg) diets resulted in heavier lambs at birth (*p =* 0.015, Table 6) when compared to the CTL group (2.64 kg). At weaning, animals from the ME treatment (16.60 kg) continued to be heavier than the others (*p =* 0.009).

The effect of the type of birth was observed on the BW of the lambs at both times evaluated (*p <* 0.0001), as single gestation ewes gave birth and weaned heavier lambs (Table 6). The sex of the lamb influenced BW at birth, with male lambs being heavier at birth (*p <* 0.0001), but males and females showed similar BW at weaning (*p >* 0.05). The birth order showed a significant effect (*p <* 0.0001) on birth weight, with lambs born to nulliparous females being lighter. However, ewes with more than 2 births weaned lighter lambs (*p =* 0.004). The analysis of the interaction of treatment with the type of birth showed a difference (*p =* 0.013), that being that lambs from single birth and MECP were the lightest. The interaction between treatment and sex of lambs showed higher earning capacity at weaning (*p =* 0.022) for males born to females in the ME group. The maternal ME diet also positively impacted the performance of male lambs at weaning.

Production efficiency at lambing and weaning were not influenced (*p >* 0.05) by treatments but suffered the effect of the type of birth, being more efficient (*p =* 0.006) for twin lambs (Table 7). Production efficiency at parturition was higher in ewes that gestated males (*p =* 0.039). There was no significant effect of birth order on production efficiency at lambing or weaning (*p >* 0.05). Analysis of the treatment interaction with the type of birth showed differences in production efficiency when dietary treatments were applied to twin-pregnant ewes, with both the ME and MECP diets being superior to the CTL group. This effect was consistent both in the immediate postpartum analysis (*p =* 0.004) and at weaning (*p =* 0.008). The interaction between treatment and lamb sex (*p =* 0.027) showed that the ME and MECP treatments were superior to CTL in raising the efficiency of ewes pregnant with male lambs.

### 3.5. Behavior

The behavior of lambs born from ME and MECP ewes was different from the control. There was a 3.5 min decrease in variability for head shaking when lambs born to ewes in the ME group were compared to the CTL group. Lambs in the ME group reduced the time required to stay on their knees by 36 s. Lambs in the MECP group had reduced the time to get on their knees by 36 s and reduced the time required to try to stand up by 42 s.

ME and MECP dietary treatments significantly reduced latency to stand by 5.3 min (*p <* 0.0001) and 4.1 min (*p =* 0.005), respectively, relative to CTL. Maternal overnutrition reduced the latency to reach the udder in lambs in the ME and MECP groups by 5.9 and 4.8 min, respectively (*p <* 0.05). There was a reduction in variability for standing of 2.0 and 2.1 min for lambs born to ewes in the ME and MECP treatments compared to lambs born to CTL ewes, respectively (*p ≤* 0.05). Additionally, variability for arrival at the udder was reduced by 2.5 min for lambs in the MECP group relative to lambs born to CTL ewes (*p ≤* 0.05).

Supplementation during gestation increased latency to first feed by 9.3 and 1.9 min for lambs in the ME and MECP treatments, respectively, relative to CTL, with opposite trends in variability (*p <* 0.0001). There was a 2.0 min reduction and 2.1 min increase in variability for first suckling for lambs born to ewes from ME and MECP treatments relative to lambs born to CTL ewes, respectively. No difference was observed between treatments for vitality score, which averaged 3.42, 3.64, and 3.39 for CTL, ME, and MECP lambs, respectively. There was also no significant variability for this characteristic (*p >* 0.05), regardless of the dietary treatment applied.

## 4. Discussion

This study contributes new knowledge on the reproductive performance of hair sheep ewes, their placental morphometry, endocrine functions, and behavior of newborn lambs after exposure of pregnant ewes to diets formulated to provide more energy or energy/protein than recommended for the prenatal period. Such knowledge may assist in the development of nutritional management strategies during the reproductive season that can be applied to increase productivity in sheep flocks.

### 4.1. Ewe’s Performance

The higher BW of the ewes in the MECP group before and after lambing is explained by the increased energy and protein concentration of the diet, indicating that the higher nutrient intake favored the females until puerperium and improved their performance indicators. As the diets in the prepartum period did not influence the BW of the ewes at weaning, the overnutrition did not show a positive residual effect in the period from lambing to weaning, demonstrating that nutrition after parturition is a primordial factor for the maintenance of the BW of the ewes in this phase.

The dietary treatment did not influence the body condition score of the ewes in the postpartum period. The ewes maintained a BCS above 3.0 at lambing and weaning, in line with the recommendation that ewes have a BCS of 3.0 to 3.5 in late gestation and early lactation and 2.5 towards the end of lactation [28].

### 4.2. Hormone Concentration

The higher thyroid activity expressed in ewes that received the experimental diets can be explained by the greater nutrient input and quality of the feed ingested in the prepartum period. Overfeeding results in changes in serum T3 levels, as the hypothalamic–pituitary–thyroid axis plays a key role in regulating energy homeostasis [29]. Our results corroborate previous findings that report higher plasma concentrations of T3 in adult sheep fed with diets higher in energy and protein [30]. Thyroid hormones also increase concentrations of somatotropin [31], which, in turn, is important for the growth and development of the mammary gland and milk production [32]. It is possible that this is the reason why both ME and MECP elevated T3 secretion at parturition, especially in younger ewes, from nulliparous to those with three deliveries.

The distinct elevation of cortisol observed in both groups of overfed females may have a dietary origin and be related to dyslipidemia induced by the high-energy diets tested. Dyslipidemia leads to greater accumulation of visceral fat (imperceptible to the visual assessment of body condition score in sheep), the development of insulin resistance, and greater assimilation of ingested triglycerides [33]. Considering the birth order, the elevation of cortisol recorded in nulliparous and primiparous ewes is explained by the fact that parturition is an event unknown or minimally known by less experienced females, possibly associated with the memory of sensations such as tension, discomfort, or pain. The BW of the ewes at lambing may have contributed to this response, as ewes with four or more pregnancies, which had higher weights at parturition than the others, had lower cortisol concentrations, which may be related to their greater physical complexion and ability at birth.

### 4.3. Placental Characteristics

Dietary treatments in the last third of gestation did not influence placental weight, but this result is interesting because larger placentas demand more nutrients for their own genesis, to the detriment of the fetus [18]. Conversely, higher placental efficiency was observed for ME ewes than CTL ewes. These results reinforce the importance of adequate nutrition in the prepartum period, especially in the last third of gestation when fetal growth is greatest. It has been shown that ewes’ placental efficiency is linked to changes in capillary vessel density [6] and that nutrition can alter blood flow to the placenta [34]. Therefore, the higher nutrient density offered to the ewes may favor fetal growth through increased nutrient transport through the placenta due to increased blood flow between the maternal and fetal circulations.

ME supplementation in the last third of gestation led to a significant reduction in total cotyledon weight. There is a reduction in total fetal cotyledon weight in ewes that had higher intake at the end of the first trimester of gestation [35]. In the present study, the size and average number of cotyledons were not influenced by the higher nutritional intake. A possible explanation for the lack of change in these cotyledon characteristics or even in placental weight may be related to the time window used in this study, which may have been short and/or late, and therefore insufficient to induce changes in these placental morphology characteristics. The effects of maternal nutrition on fetal size and organ development may be time-dependent, since in ewes the placental growth is completed by the end of the third month of gestation [36], peaking in weight between days 75 and 90 of gestation [37].

### 4.4. Lamb’s Performance

ME and MECP had an effect on BW of lambs after birth, corroborating previous findings that nutrition with high levels of starch increases glucose production and insulin release through the increase of propionate, which favors fetal development in the last third of gestation [38]. The BW of the lamb at birth showed an increase of 5.68% for ME and 3.40% for MECP, compared to the control animals.

Although overnutrition increased the weight of neonates, only ME lambs weaned heavier, with increases of 7.8% and 8.9% in relation to the weight of CTL and MECP lambs, respectively. Excess protein in the diet can result in energy expenditure by the animal to shed nitrogen from the body; this is called the urea cost and can result in reduced animal performance of the MECP treatment compared to ME [39]. These results are interesting because the feeding management of the ewes and lambs was the same from lambing until weaning, and also because in the experimental groups, a similar number of pregnancies by each ram of the same maternal breed was used, since the lambs’ growth could be influenced by paternal genetics [40] and/or by the effect of heterosis in case of crossbred animals.

This observed difference is presumed to indicate greater postnatal anabolic capacity induced by the ME diet. At birth, male lambs were also heavier and demonstrated greater physical complexion from prenatal life, an inherent characteristic of sexual dimorphism, which was accentuated when the ME diet was employed. Interestingly, overnutrition did not favor birth weight in twin births, and the ME diet further reduced birth weight in single lambs. This particular finding aligns with that reported by [3], who concluded that overfeeding ewes late in gestation can reduce lamb birth weight. In addition to weight reduction, both maternal undernutrition and overnutrition can lead to intrauterine growth restriction, negative effects on animal health, increased fetal and neonatal mortality, and even decreased carcass quality [41].

Disregarding the type of birth, the treatments offered in the last third of gestation did not significantly modify the production efficiency of the ewes at lambing and weaning. This fact is due to the outstanding weight and body condition of the ewes at lambing, and the weight of the lambs at birth and weaning. However, production efficiency was higher, both at lambing and weaning, in twin births, a phenomenon previously described in the literature [42]. In fact, despite the fact that lambs of multiple births are born with lower weight, the additive effect observed in twin births is expressive when considering the total kilograms of lamb produced per ewe. Considering only twin births, overnutrition had an extremely significant effect, by increasing production efficiency both at birth and at weaning, with increases of 29 to 30% at birth and of 23 to 33% at weaning. Considering the offspring sex, the overnutrition caused significant gains in production efficiency in females that gave birth to male lambs, with increases of 28% and 36% in the production efficiency of females fed with ME and MECP, respectively.

### 4.5. Behavior

With differential nutrient intake in the maternal diet, it was observed in lambs a reduction in the time required to perform the first attitudes of self-reflection and those required for the animal to position itself in station autonomously. However, the vitality score, a characteristic that indicates the degree of consciousness and motor activity of the neonate, was not influenced by dietary treatments. A possible explanation for this finding would be the ideal birth weight, recorded in the lambs of the three groups evaluated in this study. Birth weight has an effect on motor activity in lambs [43]. Moreover, due to bioethical issues, the experimenters did not contemplate the establishment of a group in which the ewes underwent nutritional deprivation. Therefore, even the control ewes received enough energy and protein so that their lambs did not show signs of apathy or weakness at birth.

Reducing the time required by the newborn lamb to kneel, stand, and reach the udder is of great interest, as a slow behavioral progression can have negative consequences for the neonate. After birth, getting up and suckling as quickly as possible are vital attitudes to ensure lamb survival [44]. In the present study, although there was no difference in vitality, lambs born to ME and MECP ewes were quicker to stand and more quickly positioned to perform suckling. However, this behavioral advantage did not anticipate the act of suckling. The increase in latency to the first suckling can be explained by the possibility of mobilization of greater brown fat deposits in lambs born from ewes with greater nutritional intake, which would reduce the immediate craving for suckling in the immediate postpartum period. Additionally, overfed ewes had higher cortisol concentrations at birth, which directly reflects fetal cortisol production. The action of cortisol in the body is antagonistic to insulin and is therefore analogous to glucagon, acting directly on the brain and inhibiting feeding behavior in sheep [45].

## 5. Conclusions

Maternal overnutrition in the last third of gestation had positive effects on ewes’ performance. Overfed ewes presented higher concentrations of triiodothyronine and cortisol after lambing and higher placental efficiency. The higher nutritional intake in the last third of gestation increased the birth weight of the lambs, and the energy diet had an additional favorable effect, which increased the weaning weight. Overnutrition of ewes reduced the time required for the newborn lamb to rise and position itself in the station but did not influence its neonatal vitality score. The diet with energy supplementation for ewes in late gestation was positive for the lamb production system.

## Figures and Tables

**Table 2 animals-13-00103-t002:** Means of least squares (±EP) for body weight (BW) and body condition score (BCS) of ewes submitted or not to overnutrition in the last third of gestation (CTL—control; ME—energy supplementation; MECP—energy/protein supplementation). Unfolding shown when the double interaction was significant (*p <* 0.05).

Effect	BW, kg						BCS, 1–5			
	100 Days of Gestation		12 h after Lambing		90 Days of Lactation		12 h after Lambing		90 Days of Lactation	
	Mean ± EP	*p*-Value	Mean ± EP	*p*-Value	Mean ± EP	*p*-Value	Mean ± EP	*p*-Value	Mean ± EP	*p*-Value
Treatment		0.006		0.002		0.338		0.085		0.718
CTL (*n* = 43)	46.4 (±0.7) B		42.4 (±0.5) B		40.0 (±0.5) A		3.5 (±0.1) A		3.2 (±0.1) A	
ME (*n* = 44)	47.5 (±0.6) A B		43.6 (±0.5) A B		41.0 (±0.5) A		3.5 (±0.1) A		3.2 (±0.1) A	
MECP (*n* = 41)	49.3 (±0.6) A		44.9 (±0.5) A		39.6 (±0.5) A		3.6 (±0.1) A		3.2 (±0.1) A	
Type of Birth		0.103		ns		ns		ns		0.936
Single	47.1 (±0.6) A		43.3 (±0.5) ^⸶^		39.3 (±0.4) ^⸶^		3.6 (±0.1) ^⸶^		3.2 (±0.1) A	
Twin	48.4 (±0.5) A		43.8 (±0.5) ^⸶^		40.6 (±0.6) ^⸶^		3.5 (0.1) ^⸶^		3.2 (±0.1) A	
Sex of the Lamb		0.056		0.015		ns		ns		0.740
Male	48.5 (±0.5)		44.3 (±0.4) A		40.6 (±0.6) ^⸶^		3.5 (±0.1) ^⸶^		3.2 (±0.1) A	
Female	47.0 (±0.5)		42.9 (±0.4) B		39.5 (±0.5) ^⸶^		3.5 (±0.1) ^⸶^		3.2 (±0.1) A	
Birth Order		<0.0001		<0.0001		0.001		0.727		0.738
Nulliparous	45.2 (±0.9) B		41.4 (±0.7) B		39.0 (±0.7) B		3.5 (±0.1) A		3.2 (±0.1) A	
Primiparous	46.4 (±0.8) B		43.5 (±0.6) B		38.6 (±0.6) B		3.5 (±0.1) A		3.2 (±0.1) A	
2 to 3 Births	47.0 (±0.6) B		42.5 (±0.5) B		38.2 (±0.5) B		3.5 (±0.1) A		3.1 (±0.1)	
≥4 Births	52.5 (±0.7) A		47.0 (±0.6) A		45.0 (±0.6) A		3.5 (±0.1) A		3.2 (±0.1) A	
Interactions										
Treat. * TB		ns		ns		ns				
Treat. * SL		ns		ns		ns				
Treat. * BO		ns		ns		0.008		0.078		
Treatment	Nulliparous		Primiparous			2 to 3 Births			≥4 Births	
CTL (*n* = 43)	39.7 (±1.1) A a b		39.4 (±0.9) A b			37.2 (±0.9) B b			43.7 (±1.2) B a	
ME (*n* = 44)	39.0 (±1.4) A b		37.5 (±0.9) B b			38.9 (±0.9) A b			48.5 (±1.0) A a	
MECP (*n* = 41)	38.2 (±1.3) A b		39.0 (±1.1) A b			38.4 (±0.9) A b			42.9 (±1.1) B a	

(A, B) Means followed by different letters in the column differ by Tukey’s test (*p ≤* 0.05). (a, b) Means followed by different letters in the row differ by Tukey’s test (*p ≤* 0.05). ^⸶^ arithmetic mean (±EP). ns: effect not selected in the model. CTL: control diet according to [14], ME: 21% more metabolizable energy, MECP: 26% more metabolizable energy and crude protein. Treat.: treatment, TB: type of birth, SL: sex of the lamb, BO: birth order.

**Table 3 animals-13-00103-t003:** Means of least squares (±EP) for serum concentration (ng/dL) of triiodothyronine and cortisol after lambing of ewes submitted or not to overnutrition in the last third of gestation (CTL—control; ME—energy supplementation; MECP—energy/protein supplementation). Unfolding shown when the double interaction was significant (*p <* 0.05).

Effect	Triiodothyronine		Cortisol	
	Mean ± EP	*p*-Value	Mean ± EP	*p*-Value
Treatment		0.001		0.004
CTL (*n* = 43)	201.0 (±9.4) B		5.0 (±1.1) B	
ME (*n* = 44)	250.0 (±9.4) A		14.3 (±2.2) A	
MECP (*n* = 41)	274.0 (±10.4) A		18.9 (±2.9) A	
Type of Birth		ns		0.001
Single	218.2 (±9.4) ^⸶^		15.7 (±2.0) A	
Twin	249.2(±9.1) ^⸶^		8.2 (±1.2) B	
Sex of the Lamb		0.171		-
Male	234.0 (±8.3) A		15.0 (±1.6) ^⸶^	
Female	249.0 (±7.5) A		17.4 (±1.6) ^⸶^	
Birth Order		0.004		<0.0001
Nulliparous	285.0 (±13.9) A		17.2 (±3.6) A	
Primiparous	263.0 (±10.8) A		19.2 (±3.1) A	
2 to 3 Births	220.0 (±8.8) B		5.4 (±1.1) C	
≥4 Births	198.0 (±11.3) B		8.5 (±1.8) B C	
Interactions				
Treat. * TB		ns		ns
Treat. * SL		ns		ns
Treat. * BO		0.002		0.001
	Triiodothyronine			
Treatment	Nulliparous	Primiparous	2 to 3 Births	≥4 Births
CTL (*n* = 43)	230.0 (±22.3) B a	211.0 (±15.9) B a	148.0 (±15.3) C b	216.0 (±21.0) A a
ME (*n* = 44)	321.0 (±25.7) A a	275.0 (±16.3) A a b	230.0 (±15.3) B b	173.0 (±16.3) B c
MECP (*n* = 41)	303.0 (±24.2) A b a	304.0 (±22.5) A a	283.0 (±15.6) A a	207.0 (±21.0) A b
	Cortisol			
Treatment	Nulliparous	Primiparous	2 to 3 Births	≥4 Births
CTL (*n* = 43)	17.0 (±5.7) B a	11.6 (±3.8) B a	0.9 (±0.5) B b	1.2 (±0.8) B b
ME (*n* = 44)	13.2 (±5.7) B a	13.0 (±3.7) B a	14.1 (±3.5) A a	16.8 (±4.2) A a
MECP (*n* = 41)	22.2 (±7.4) A a b	40.4 (±10.5) A a	6.9 (±2.2) A b	16.2 (±5.3) A a b

(A, B) Means followed by different letters in the column differ by Tukey’s test (*p ≤* 0.05). (a, b) Means followed by different letters in the row differ by Tukey’s test (*p ≤* 0.05). ^⸶^ arithmetic mean (±EP). ns: effect not selected in the model. CTL: control diet according to [14], ME: 21% more metabolizable energy, MECP: 26% more metabolizable energy and crude protein. Treat.: treatment, TB: type of birth, SL: sex of the lamb, BO: birth order.

**Table 4 animals-13-00103-t004:** Means of least squares (±EP) for placental development characteristics of ewes submitted or not submitted to overnutrition in the last third of gestation (CTL—control; ME—energy supplementation; MECP—energy/protein supplementation).

	Weight of the Placenta (g)		Placental Efficiency	
	Mean ± EP	*p*-Value	Mean ± EP	*p*-Value
Treatment		0.181		0.036
CTL (*n* = 43)	383.0 (±14.5) A		10.4 (±0.41) B	
ME (*n* = 44)	350.0 (±14.4) A		11.7 (±0.39) A	
MECP (*n* = 41)	356.0 (±14.0) A		10.8 (±0.41) A B	
Type of Birth		<0.0001		0.019
Single	299.0 (±12.1) B		10.4 (±0.34) B	
Twin	428.0 (±11.5) A		11.5 (±0.32) A	
Sex of the Lamb		0.604		0.016
Male	374.0 (±12.0) A		11.5 (±0.33) A	
Female	353.0 (±11.5) A		10.4 (±0.33) B	
Birth Order		0.068		0.525
Nulliparous	325.0 (±20.2) A		10.9 (±0.55) A	
Primiparous	361.0 (±17.2) A		11.4 (±0.49) A	
2 to 3 Births	375.0 (±14.5) A		11.0 (±0.39) A	
≥4 Births	392.0 (±15.5) A		10.5 (±0.44) A	
Interactions				
Treat. * TB		-		-
Treat. * SL		0.137		-
Treat. * BO		-		0.078

(A, B) Means followed by different letters in the column differ by Tukey’s test (*p ≤* 0.05). Ns: effect not selected in the model. CTL: control diet according to [14], ME: 21% more metabolizable energy, MECP: 26% more metabolizable energy and crude protein. Treat.: treatment, TB: type of birth, SL: sex of the lamb, BO: birth order.

**Table 5 animals-13-00103-t005:** Means of least squares (±EP) of characteristics of placental cotyledons of ewes submitted or not to overnutrition in the last third of gestation (CTL—control; ME—energy supplementation; MECP—energy/protein supplementation). Unfolding shown when the double interaction was significant (*p <* 0.05).

	Total Weight (g)		Diameter (mm)		Total Number	
	Mean ± EP	*p*-Value	Mean ± EP	*p*-Value	Mean ± EP	*p*-Value
Treatment		0.011		-		0.774
CTL (*n* = 43)	91.4 (±3.3) A		20.8 (±0.5) ^⸶^		71.6 (±1.4) A	
ME (*n* = 44)	81.5 (±3.3) B		19.8 (±0.4) ^⸶^		67.4 (±1.3) A	
MECP (*n* = 41)	87.5 (±3.2) A B		20.3 (±0.5) ^⸶^		68.6 (±1.4) A	
Type of Birth		<0.0001		0.001		<0.0001
Single	68.9 (±2.8) B		19.2 (±0.4) B		66.5 (±1.2) B	
Twin	104.7 (±2.6) A		21.2 (±0.3) A		71.9 (±1.1) A	
Sex of the Lamb		0.028		0.039		-
Male	89.3 (±2.8) A		20.7 (±0.4) A		69.9 (±1.2) ^⸶^	
Female	84.3 (±2.6) B		19.7 (±0.3) B		69.4 (±1.1) ^⸶^	
Birth Order		0.012		0.011		0.366
Nulliparous	78.5 (±4.7) B		20.2 (±0.6) A		69.2 (±1.8) A	
Primiparous	86.0 (±3.9) A B		19.1 (±0.5) B		71.5 (±1.7) A	
2 to 3 Births	85.6 (±3.3) A B		20.0 (±0.4) A		67.3 (±1.3) A	
≥4 Births	97.1 (±3.5) A		21.5 (±0.5) A		68.8 (±1.4) A	
Interactions						
Treat. * TB		-		-		0.005
Treat. * SL		0.069		-		-
Treat. * BO		-		-		0.007
Treatment	Type of Birth					
	Single			Twin		
CTL (*n* = 43)	66.1 (±2.1) B b			77.7 (±1.9) A a		
ME (*n* = 44)	68.1 (±1.8) A a			66.7 (±2.0) B a		
MECP (*n* = 41)	65.4 (±2.0) B b			71.9 (±1.8) A b a		
Treatment	Birth Order					
	Nulliparous	Primiparous		2 to 3 Births	≥4 Births	
CTL (*n* = 43)	73.4 (±3.0) A a	74.8 (±3.3) A a		68.8 (±2.3) A a	69.7 (±2.6) B a	
ME (*n* = 44)	61.8 (±2.8) B b	69.5 (±2.9) B a b		65.0 (±2.3) B b	74.0 (±2.5) A a	
MECP (*n* = 41)	73.1 (±3.5) A a	70.4 (±2.6) B a		68.1 (±2.1) A a	63.2 (±2.4) C a	

(A, B) Means followed by different letters in the column differ by Tukey’s test (*p ≤* 0.05). (a, b) Means followed by different letters in the row differ by Tukey’s test (*p ≤* 0.05). ^⸶^ arithmetic mean (±EP). ns: effect not selected in the model. CTL: control diet according to [14], ME: 21% more metabolizable energy, MECP: 26% more metabolizable energy and crude protein. Treat.: treatment, TB: type of birth, SL: sex of the lamb, BO: birth order.

**Table 6 animals-13-00103-t006:** Means of least squares (±EP) for the weight of lambs (kg) born from ewes submitted or not to overnutrition in the last third of gestation (CTL—control; ME—energy supplementation; MECP—energy/protein supplementation). Unfolding shown when the double interaction was significant (*p <* 0.05).

Effect	Weight at Birth		Weight at Weaning	
	Mean ± EP	*p*-Value	Mean ± EP	*p*-Value
Treatment		0.015		0.009
CTL (*n* = 62)	2.64 (±0.05) B		15.50 (±0.46) B	
ME (*n* = 59)	2.79 (±0.05) A		16.60 (±0.46) A	
MECP (*n* = 54)	2.73 (±0.05) A		15.50 (±0.47) B	
Type of Birth		<0.0001		<0.0001
Single	3.01 (±0.04) A		17.20 (±0.42) A	
Twin	2.43 (±0.04) B		14.50 (±0.37) B	
Sex of the Lamb		<0.0001		0.265
Male	2.86 (±0.04) A		16.30 (±0.42) A	
Female	2.58 (±0.03) B		15.40 (±0.36) A	
Birth Order		<0.0001		0.004
Nulliparous	2.39 (±0.07) B		17.30 (±0.90) A	
Primiparous	2.81 (±0.05) A		16.40 (±0.49) A	
2 to 3 Births	2.79 (±0.04) A		14.50 (±0.41) B	
≥4 Births	2.90 (±0.05) A		15.20 (±0.48) A B	
Interactions				
Treat. * TB	-	0.013	-	
Treat. * SL	-		-	0.022
Treat. * BO	-		-	
	Weight at Birth			
Treatment	Type of Birth			
	Single		Twin	
CTL (*n* = 62)	2.95 (±0.07) B a		2.34 (±0.06) B b	
ME (*n* = 59)	3.18 (±0.07) A a		2.40 (±0.06) B b	
MECP (*n* = 54)	2.92 (±0.07) B a		2.55 (±0.06) A b	
	Weight at Weaning			
Treatment	Sex of the Lamb			
	Male		Female	
CTL (*n* = 62)	15.00 (±0.65) B b		16.00 (±0.63) B a	
ME (*n* = 59)	17.80 (±0.69) A a		15.40 (±0.55) B b	
MECP (*n* = 54)	16.10 (±0.67) A B a		14.80 (±0.65) A b	

(A, B) Means followed by different letters in the column differ by Tukey’s test (*p ≤* 0.05). (a, b) Means followed by different letters in the row differ by Tukey’s test (*p ≤* 0.05). ns: effect not selected in the model. CTL: control diet according to [14], ME: 21% more metabolizable energy, MECP: 26% more metabolizable energy and crude protein. Treat.: treatment, TB: type of birth, SL: sex of the lamb, BO: birth order.

**Table 7 animals-13-00103-t007:** Means of least squares (±EP) for production efficiency at lambing and weaning of ewes submitted or not to overnutrition in the last third of gestation (CTL—control; ME—energy supplementation; MECP—energy/protein supplementation). Unfolding shown when the double interaction was significant (*p <* 0.05).

	At Birth		At Weaning	
	Mean ± EP	*p*-value	Mean ± EP	*p*-value
Treatment		0.161		0.196
CTL (*n* = 43)	1.03 (±0.04)		1.18 (±0.45)	
ME (*n* = 44)	1.25 (±0.04)		1.37 (±0.05)	
MECP (*n* = 41)	1.21 (±0.04)		1.40 (±0.05)	
Type of Birth		0.006		0.008
Single	0.96 (±0.04) B		1.07 (±0.04) B	
Twin	1.37 (±0.04) A		1.56 (±0.04) A	
Sex of the Lamb		0.039		0.163
Male	1.20 (±0.03) A		1.35 (±0.43)	
Female	1.12 (±0.03) B		1.28 (±0.36)	
Birth Order		-		-
Nulliparous	1.13 (±0.05) ^⸶^		1.44 (±0.18) ^⸶^	
Primiparous	1.13 (±0.05) ^⸶^		1.24 (±0.06) ^⸶^	
2 to 3 Births	1.01 (±0.04) ^⸶^		1.21 (±0.06) ^⸶^	
≥4 Births	1.13 (±0.07) ^⸶^		1.26 (±0.07) ^⸶^	
Interactions				
Treat. * TB		0.004		0.008
Treat. * SL		-		0.027
Treat. * BO		-		-
	Production Efficiency at Birth			
Treatment	Type of Birth			
	Single		Twin	
CTL (*n* = 43)	0.93 (±0.05) B b		1.14 (±0.05) B a	
ME (*n* = 44)	1.02 (±0.04) A b		1.47 (±0.05) A a	
MECP (*n* = 41)	0.92 (±0.04) B b		1.49 (±0.06) A a	
	Production Efficiency at Weaning			
Treatment	Type of Birth			
	Single		Twin	
CTL (*n* = 43)	1.05 (±0.07) B b		1.31 (±0.07) B a	
ME (*n* = 44)	1.12 (±0.05) A b		1.62 (±0.07) A a	
MECP (*n* = 41)	1.04 (±0.06) B b		1.75 (±0.09) A a	
Treatment	Sex of the Lamb			
	Male		Female	
CTL (*n* = 43)	1.11 (±0.07) B b		1.25 (±0.06) B a	
ME (*n* = 44)	1.43 (±0.08) A a		1.31 (±0.06) A b	
MECP (*n* = 41)	1.51 (±0.07) A b		1.28 (±0.07) B a	

(A, B) Means followed by different letters in the column differ by Tukey’s test (*p ≤* 0.05). (a, b) Means followed by different letters in the row differ by Tukey’s test (*p ≤* 0.05). ^⸶^ arithmetic mean (±EP). ns: effect not selected in the model. CTL: control diet according to [14], ME: 21% more metabolizable energy, MECP: 26% more metabolizable energy and crude protein. Treat.: treatment, TB: type of birth, SL: sex of the lamb, BO: birth order. Production efficiency at birth and at weaning were calculated by the ratio between lamb weight at weaning and the ewe’s metabolic weight at lambing and at weaning, respectively. Metabolic weight = (Body Weight)^0.75^.

## Data Availability

The research data and other artefacts supporting the results reported in the paper can be found in the Embrapa’s official repository. Access to original dataset may be requested to the institution and is subjected to institutional approval.

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
