# Peer review of "Overnutrition of Ewe in Late Gestation and the Impact on Placental Efficiency and Lamb’s Performance"

_animals, 2022, doi:10.3390/ani13010103_

Round 1
Reviewer 1 Report
1. Please explain the steps and methods of synchronous estrus treatment in detail.
2. Why not measure progesterone and estrogen levels?
3. Increasing the energy and protein levels of the diet will increase the birth weight of the fetus and may result in dystocia of the ewes. Is the ease of birth measured and recorded?
4. Increasing the energy and protein levels in the diet not only affects the growth and development of lambs, but also may affect the condition of ewes after delivery. Whether the weight and estrus of the ewes have been recorded and counted after delivery?
5. Different parity may interfere with the reproductive performance of ewes. Do you compare and analyze the parity factors of ewes?
6. The lambs with more than four fetuses may have an impact on the growth and development of the lambs. How to ensure that the lamb has enough milk nutrition intake, without affecting the growth and development of the lamb? Are there any recorded data on lambs' weaning survival?
7. Statistics analysis are too complicated to understand, please simplify.
8. The increase of energy and protein level in the diet of ewes in late gestation can significantly increase the BW of lambs, but the weaning weight of MECP group is significantly lower than that of ME group. The MECP group has high energy and high protein diet, so what is the effect of adding high protein on the weight gain of lambs during lactation? Is it antagonistic?
Author Response
Responses to Reviewer #1
We thank the referee for her/his thoughtful suggestions. We have taken them into account when revising the manuscript.
The corrections were incorporated into the new version of the manuscript and all changes were highlighted.
All amendments are referred considering the numbered lines of the current version of the manuscript.
In this section we present the answers to the referee.
Reviewer 1
Comments and Suggestions for Authors
(x) Moderate English changes required
Authors´ Resp: Thank you for the observation about moderate English changes required. The manuscript was revised by a certified English translator and a proof-reading declaration was emitted.
- Please explain the steps and methods of synchronous estrus treatment in detail.
Authors´ Resp: Thank you for the suggestion. The estrus synchronization protocol was detailed (lines 100-101).
- Why not measure progesterone and estrogen levels?
Authors´ Resp: Thank you for this question. This analysis was in the original experimental design. However, since the resources were limited, we opted to focus on variables more strictly related to the performance of ewes and lambs.
- Increasing the energy and protein levels of the diet will increase the birth weight of the fetus and may result in dystocia of the ewes. Is the ease of birth measured and recorded?
Authors´ Resp: Thank you for this question. As an increase in the birth weight was expected, all parturitions were closely observed, including deliveries that occurred in the early hours. Veterinary assistance was on standby in case of for any intercurrence; we inserted this information in the manuscript (lines 169-169). The ease of birth was measured and data will be presented and discussed in a complementary article.
- Increasing the energy and protein levels in the diet not only affects the growth and development of lambs, but also may affect the condition of ewes after delivery. Whether the weight and estrus of the ewes have been recorded and counted after delivery?
Authors´ Resp: Thank you for the question. The weight of ewes was monitored immediately after lambing and in the weaning, as presented in Table 2 (line 264) and in the Discussion section. Intermediary weigh controls were not performed. Estrus was not controlled after lambing as the all-day nursing regime was adopted and it is involved on the physiological postpartum anestrus mechanisms.
- Different parity may interfere with the reproductive performance of ewes. Do you compare and analyze the parity factors of ewes?
Authors´ Resp: Yes, we did. The parity of ewes was included in all statistical models adopted in the experiment. It was represented as “birth order” (nulliparous, primiparous, multiparous with 2 or 3 births, and multiparous with 4 or more births). The effect of parity order and its respective interactions were presented in Tables 2-7.
- The lambs with more than four fetuses may have an impact on the growth and development of the lambs. How to ensure that the lamb has enough milk nutrition intake, without affecting the growth and development of the lamb? Are there any recorded data on lambs' weaning survival?
Authors´ Resp: As only ewes with single or twin gestations (that ones containing two fetuses) were used in this experiment, the effects of births with three or more lambs did not occur (lines 107-109). The strict veterinary monitoring of the ewes and lambs guaranteed no animal loss during the experiment.
- Statistics analysis are too complicated to understand, please simplify.
Authors´ Resp: Thank you for this recommendation. The statistical description was simplified (lines 222-236).
- The increase of energy and protein level in the diet of ewes in late gestation can significantly increase the BW of lambs, but the weaning weight of MECP group is significantly lower than that of ME group. The MECP group has high energy and high protein diet, so what is the effect of adding high protein on the weight gain of lambs during lactation? Is it antagonistic?
Authors´ Resp: The amount of protein in the MECP diet may have been above the animal's requirement and, therefore, the ewes had to expend metabolic energy to excrete excess nitrogen from the diet. This may have been the cause of the worse performance of MECP compared to ME treatment. An explanation was inserted in the manuscript (lines 457-459) and the respective reference was inserted in the list.
Reviewer 2 Report
The authors investigated the effect of the energy/protein supplementation in the ewe diet on the maternal placental and endocrine characteristics, and the behavior of neonatal lambs and productive performance until weaning. In general, the organization and the structure of the article are satisfactory, however, the authors should follow the journal instructions for authors, while preparing the manuscript (simple summary section, reference style etc.). My suggestion is to accept after major revisions.
The other comments have been listed below:
1. The “P” should be italicized, check it throughout the manuscript;
2. This journal does not require the “Highlights”, instead the authors should add the “Simple Summary” section;
3. The weight and body morphometric traits of normally developed and well-fed animals might be influenced by the individual hereditary characteristics of the parents. Therefore, the authors should provide additional information about the Morada Nova breed in the Introduction section.
4. Add the Ethics statement in the “Materials and Methods” section;
5. The body weight and body morphometric traits of animals might be influenced by the genetic factors. The authors requested to discuss about this issue;
6. Use the template of the present journal, also for references style.
Author Response
Responses to Reviewer #2
We thank the referee for her/his thoughtful suggestions. We have taken them into account when revising the manuscript.
The corrections were incorporated into the new version of the manuscript and all changes were highlighted.
All amendments are referred considering the numbered lines of the current version of the manuscript.
In this section we present the answers to the referee.
Reviewer 2
Comments and Suggestions for Authors
The authors investigated the effect of the energy/protein supplementation in the ewe diet on the maternal placental and endocrine characteristics, and the behavior of neonatal lambs and productive performance until weaning. In general, the organization and the structure of the article are satisfactory, however, the authors should follow the journal instructions for authors, while preparing the manuscript (simple summary section, reference style etc.). My suggestion is to accept after major revisions.
The other comments have been listed below:
- The “P” should be italicized, check it throughout the manuscript;
Authors´ Resp: “P” was italicized throughout the text.
- This journal does not require the “Highlights”, instead the authors should add the “Simple Summary” section;
Authors´ Resp: Thank you for this observation. The highlights were suppressed and a “Simple Summary” was inserted (lines 26-38).
- The weight and body morphometric traits of normally developed and well-fed animals might be influenced by the individual hereditary characteristics of the parents. Therefore, the authors should provide additional information about the Morada Nova breed in the Introduction section.
Authors´ Resp: Thank you for this suggestion. Even the animals are part from an experimental flock in which frame size and purebred features are as homogeneous as possible, we considered the parental effect. Thus, similar numbers of pregnancies of the rams were assigned to the treatments. In addition, we provided more information about the Morada Nova breed in the Introduction section (lines 64-67) and the respective reference was inserted in the list.
- Add the Ethics statement in the “Materials and Methods” section;
Authors´ Resp: The “Ethics statement” was inserted in the “Materials and Methods” section (lines 91-96), and the “Institutional Review Board Statement” was maintained, but in a synthetized form to avoid redundancy (lines 543-544).
- The body weight and body morphometric traits of animals might be influenced by the genetic factors. The authors requested to discuss about this issue;
Authors´ Resp: Thank you for this suggestion. This issue was inserted in the Discussion (lines 459-464) and the respective reference was inserted in the list.
- Use the template of the present journal, also for references style.
Authors´ Resp: Thank you for this comment. The manuscript was configured according the Animal’s Guide for the Authors, including citations and the references.
Author Response
Responses to Reviewer #3
We thank the referee for her/his thoughtful suggestions. We have taken them into account when revising the manuscript.
The corrections were incorporated into the new version of the manuscript and all changes were highlighted.
All amendments are referred considering the numbered lines of the current version of the manuscript.
In this section we present the answers to the referee.
Reviewer 3
(x) English language and style are fine/minor spell check required
Authors´ Resp: Thank you for the observation about English check requirement. The manuscript was revised by a certified English translator and a proof-reading declaration was emitted.
Summary: This manuscript adds to the growing base of knowledge regarding fetal programming, specifically overnutrition during the last trimester of gestation. The authors did a large amount of data collection and analysis to make this a versatile and widely applicable paper. It is also of note to point out that the researchers used hair sheep, whereas a number of other ovine fetal programming studies have utilized wool sheep.
General comments about the article:
How was the calculation of 17% or 20% over NRC values done, as based upon the presented information I calculated them (from Table 1) to be 21% or 26% over the control which is equivalent to the NRC values, correct?
Authors´ Resp: The correct percentage is 21% in the ME treatment and 26% in the MECP treatment. It has already been corrected in the text (lines 45; 118-119) and tables. Thank you very much for the observation.
The “unfolded” portions of Table 2 through Table 7 are cumbersome and somewhat complicated to understand at first glance. I wonder if there is a better way to present them which may be easier for a reader to identify the important points you are trying to convey. Also, on those same tables (2 – 7) the distinction between capital letters (A,B,C) and noncapital letters (a,b,c) is not well defined.
Authors´ Resp: Thank you for this observation. We agree with the reviewer that the Tables are complex and their interpretation require an extra attention from the reader. However, the experimental design and the statistical models adopted leave us few options to summarize the data. We tried to move all this information to graphics but the visual result was unsatisfactory and less informative. So, we have chosen to keep the presentation in tables. In addition, the capital letters and noncapital letters was reviewed in all Tables.
How specifically was dry matter intake estimated from Equation 1 which gives you BW at lambing?
Authors´ Resp: Thank you for the question. The real body weight of the animals was used in the equation, as well as that of the lambs to estimate the consumption of dry matter. The research team has used this flock in other previous experiments. Therefore, there is a record track of the weight control of the animals in pregnancy and lactation. Prior to the experiment, this knowledge was used in the equation and briefly the consumption of dry matter that the animal would have. However, the consumption shown in Table 1 was based on the body weight of ewes and lambs throughout the experiment.
General comments about the review: Most previous research in the programming area looks at undernutrition, so I commend the authors for finding the overnutrition studies which apply to their objectives and project.
Authors´ Resp: Thank you for this relevant observation. We opted not to include an undernourished experimental group due to bioethical concerns. As the work was carried out in a governmental experimental field, we followed the federal normative to keep experimental animals that includes regular nutrition of pregnant females. However, we agree that undernutrition is a problem frequently encountered in many properties. We inserted an argument about overnutrition studies in the Discussion Section (lines 457-459; 471-474) and the respective reference was inserted in the list.
Specific comments:
On Line 37, “effectson” should be changed to “effects on”.
Authors´ Resp: The amendment was done (line 41).
On Line 67, remove “the” after during and change “raise” to “rise”.
Authors´ Resp: The amendment was done (line 75)
On Line 76, change the “the” after of to “this”.
Authors´ Resp: The amendment was done (line 83).
Line 85 and 86, change the beginning of the sentence to “Utilizing 134 Morada Nova hair sheep ewes,”.
Authors´ Resp: The amendment was done (lines 98-99).
The sentence beginning on Line 91 and ending Line 93, I would rephrase to “A total of 128 ewes were confirmed pregnant and utilized for the experiment through lambing, with 172 lambs resulting for use in the study”.
Authors´ Resp: Thank you for the suggestion. The phrase was changed (lines 106-109).
On Line 102, add “only” after 17% more ME.
Authors´ Resp: Thank you for the suggestion. The amendment was done (line 118).
On Line 141, change “observed” to “monitored”.
Authors´ Resp: The amendment was done (line 166).
Also on Line 141, should “specific” be “progressive”?
Authors´ Resp: Thank you for the suggestion. The term was changed (line 166).
On Line 145, change “up to” to “within” and “after” to “of”.
Authors´ Resp: The amendment was done (line 173).
For Line 146, add “scales” after “cage type”.
Authors´ Resp: The amendment was done (line 174).
On Line 150, add “a” after “aid of”.
Authors´ Resp: The amendment was done (line 178).
On Line 151, is 1 g correct or should it be 1 kg?
Resp: Thank you for the question. It is correct.
On Line 160, change “T3 levels” to “Levels of T3”.
Authors´ Resp: The amendment was done (line 188).
On Line 207, change “calving” to “lambing”.
Authors´ Resp: The term was changed (line 229).
In Line 221, change “MECP resulted” to “The MECP treatment resulted”.
Authors´ Resp: The amendment was done (line 240).
On Line 223, remove “with values of”.
Authors´ Resp: The expression was removed.
For Table 2, is the Treat*BO interaction unfolded in the table or not and if not why?
Authors´ Resp: The mentioned interaction is shown in the Table 2.
In Table 4, Placental Efficiency has a symbol which is not defined at the bottom of the table.
Resp: Thank you for the observation. The symbol was removed.
On Line 281, change “calving” to “lambing”.
Authors´ Resp: The term was changed (lines 300).
On Lines 153, 232, and 383, change “Hormone dosage” to “Hormone concentration” as these were not given to the animal exogenously.
Authors´ Resp: The amendments were done (lines 181; 251; 404).
Round 2
